# The characteristics of bacteremia among patients with acute febrile illness requiring hospitalization in Indonesia

Pratiwi Soedarmono[1], Aly Diana[2,3], Patricia Tauran[4], Dewi Lokida[5], Abu Tholib Aman[6], Bachti Alisjahbana[7], Dona Arlinda[8], Emiliana Tjitra[8], Herman Kosasih[2]*, Ketut Tuti Parwati Merati[9], Mansyur Arif[4], Muhammad Hussein Gasem[10], Nugroho Harry Susanto[2], Nurhayati Lukman[2], Retna Indah Sugiyono[2], Usman Hadi[11], Vivi Lisdawati[12], Karine G. Fouth Tchos[13], Aaron Neal[13], Muhammad Karyana[2,8]

1 Faculty of Medicine, Universitas Indonesia/ Dr. Cipto Mangunkusumo Hospital, Jakarta, Indonesia,
2 Indonesia Research Partnership on Infectious Disease (INA-RESPOND), Jakarta, Indonesia,
3 Department of Public Health, Faculty of Medicine, Universitas Padjadjaran, Sumedang, Indonesia,
4 Faculty of Medicine, Universitas Hasanuddin/ Dr. Wahidin Sudirohusodo Hospital, Makassar, Indonesia,
5 Tangerang District Hospital, Tangerang, Banten, Indonesia, 6 Faculty of Medicine, Public Heath, and Nursing, Universitas Gadjah Mada/ Dr. Sardjito Hospital, Yogyakarta, Indonesia, 7 Department of Internal Medicine, Faculty of Medicine, Universitas Padjadjaran/ Dr Hasan Sadikin Hospital, Bandung, Indonesia,
8 National Institute of Health Research and Development (NIHRD), Ministry of Health Republic of Indonesia, Jakarta, Indonesia, 9 Faculty of Medicine, Udayana University, Sanglah General Hospital, Denpasar, Bali, Indonesia, 10 Faculty of Medicine, Diponegoro University/ Dr. Kariadi Hospital, Semarang, Indonesia, 11 Faculty of Medicine, Universitas Airlangga/ Dr. Soetomo Hospital, Surabaya, Indonesia, 12 Sulianti Saroso Infectious Disease Hospital, Jakarta, Indonesia, 13 National Institute of Allergy and Infectious Disease (NIAID), National Institutes of Health, Bethesda, Maryland, United States of America

* hkosasih@ina-respond.net

**Data Availability Statement:** All relevant data are within the paper and its Supporting Information files.

## Abstract

Blood culturing remains the "gold standard" for bloodstream infection (BSI) diagnosis, but the method is inaccessible to many developing countries due to high costs and insufficient resources. To better understand the utility of blood cultures among patients in Indonesia, a country where blood cultures are not routinely performed, we evaluated data from a previous cohort study that included blood cultures for all participants. An acute febrile illness study was conducted from July 2013 to June 2016 at eight major hospitals in seven provincial capitals in Indonesia. All participants presented with a fever, and two-sided aerobic blood cultures were performed within 48 hours of hospital admission. Positive cultures were further assessed for antimicrobial resistance (AMR) patterns. Specimens from participants with negative culture results were screened by advanced molecular and serological methods for evidence of causal pathogens. Blood cultures were performed for 1,459 of 1,464 participants, and the 70.6% (1,030) participants that were negative by dengue NS1 antigen test were included in further analysis. Bacteremia was observed in 8.9% (92) participants, with the most frequent pathogens being *Salmonella enterica serovar* Typhi (41) and Paratyphi A (10), *Escherichia coli* (14), and *Staphylococcus aureus* (10). Two *S*. Paratyphi A cases had evidence of AMR, and several *E. coli* cases were multidrug resistant (42.9%, 6/14) or mono-resistant (14.3%, 2/14). Culture contamination was observed in 3.6% (37) cases. Molecular and serological assays identified etiological agents in participants having negative cultures,

**Funding:** This project has been funded in whole or in part with MOH Indonesia and Federal funds from the National Institute of Allergy and Infectious Diseases, National Institutes of Health, under contract Nos. HHSN261200800001E and HHSN261201500003I. The content of this publication does not necessarily reflect the views or policies of the Department of Health and Human Services, nor does mention of trade names, commercial products, or organizations imply endorsement by the U.S. Government.

**Competing interests:** The authors have declared that no competing interests exist.

with 23.1% to 90% of cases being missed by blood cultures. Blood cultures are a valuable diagnostic tool for hospitalized patients presenting with fever. In Indonesia, pre-screening patients for the most common viral infections, such as dengue, influenza, and chikungunya viruses, would maximize the benefit to the patient while also conserving resources. Blood cultures should also be supplemented with advanced laboratory tests when available.

## Introduction

Bloodstream infections (BSI) [1] are a significant cause of morbidity and mortality in both developing and developed countries [2–4]. The "gold standard" method for BSI diagnosis remains blood culturing [5–7], a straightforward laboratory technique that is inaccessible to many developing countries due to high costs and insufficient resources. Blood cultures provide both definitive microbiological evidence of infection and serve as a crucial tool to monitor the serious global health threat of antimicrobial resistance (AMR) [8]. The threat of AMR further exacerbates the burden felt in countries without routine access to this diagnostic method, including in Indonesia, and allows AMR to continue threatening populations worldwide. The early and accurate identification of causative microorganisms and their susceptibility to antibiotics is essential to improve patient survival and prevent emerging AMR pathogens.

Even with access to routine blood cultures, the interpretation of results can be challenging and should align with clinical observations. Bacterial growth is a consequence of the initial quantity of bacteria in the specimen, the quality of the specimen, the timing of specimen collection with clinical treatment, and the biological nature of the bacteria. Negative blood cultures alone are not definitive for diagnosis, as advanced laboratory methods often detect missed culturable organisms from the same specimen types [9, 10]. Routine analysis of specimens can be impacted by contamination from the environment of the patient [11, 12]. In most settings, only 5 to 13% of blood cultures will become positive, and of those, 20–56% result from contamination [7, 13–16].

In Indonesia, acute febrile illness resulting from BSIs remains a common cause of hospitalization, morbidity, and mortality. Although infectious diseases are the primary cause of hospitalization in the country, clinicians do not routinely perform blood cultures as part of standard clinical care [17]. When clinicians perform blood cultures, generally in severely ill patients referred to tertiary care, they do not consistently use best laboratory practices [18]. Data on blood culture use, performance, and contamination rates in Indonesia remain very limited [17, 19, 20]. Consequently, data on the emergence and spread of AMR pathogens in the country is unreliable and incomplete, complicating antibiotic stewardship efforts in the region.

The epidemiology of pathogens associated with fever in Indonesia is not well understood, as public health surveillance data is limited and only a few local studies have been conducted [19, 21–26]. Among published studies, dengue virus, chikungunya virus, influenza virus, *Salmonella enterica serovar* Typhi, *Rickettsia spp.*, and *Leptospira spp.* are consistently the most common causes of acute febrile illness hospitalizations. A study in Papua from November 1997 to February 2000 enrolled 226 hospitalized patients that were negative for malaria, the majority of whom were determined to have typhoid fever (18%), leptospirosis (12%), rickettsioses (8%), and dengue fever (7%) [23]. An observational fever study in Bandung identified dengue virus in 12.4% of fever episodes, followed by *S.* Typhi (7.4%), and chikungunya virus (7.1%) [24, 26, 27]. A 2005–2006 study in Semarang found rickettsioses and leptospirosis in 7% and 10%, respectively, of 137 acute undifferentiated fever cases [21]. The parent study of

the research presented here found the most prevalent pathogens among participants at eight hospitals in 7 major cities in Indonesia to be dengue virus (27–52%), *Rickettsia spp.* (2–12%), *S.* Typhi (0.9–13%), influenza virus (2–6%), *Leptospira spp.* (0–5%), and chikungunya virus (0–4%) [19].

To better understand the utility of blood cultures among patients with acute febrile illness in Indonesia, we evaluated data from a previously published multicenter observational prospective cohort study conducted across the country [19]. Gaining insight into pathogens commonly identified by blood culture, contamination rates, AMR patterns, and disease outcomes will provide actionable evidence to support decision making for Indonesia's national blood culture testing policy.

## Methods

### Study design and sample collection

A prospective observational study enrolling febrile patients who required hospitalization was conducted by the Indonesia Research Partnership on Infectious Disease (INA-RESPOND) from July 2013 to June 2016 at eight major hospitals in seven provincial capitals in Indonesia: Dr. Cipto Mangunkusumo Hospital in Jakarta, Sulianti Saroso Infectious Disease Hospital in Jakarta, Dr. Wahidin Sudirohusodo Hospital in Makassar, Dr. Sardjito Hospital in Yogyakarta, Dr. Hasan Sadikin Hospital in Bandung, Sanglah General Hospital in Denpasar, Dr. Soetomo Hospital in Surabaya, and Dr. Kariadi Hospital, in Semarang. The full details of this study, known as AFIRE, were published previously [19]. Briefly, inclusion criteria consisted of axillary body temperature ≥38°C, ≥1 year of age, and hospitalization within the past 24 hours. Patients were excluded from the study if they had subjective fever for ≥14 days or were hospitalized in the last 3 months. Demographic, clinical, and laboratory data, including hematology results, were collected at baseline, once during days 14–28, and three months after enrollment. Blood and other biological specimens were collected at each study visit.

During the baseline visit, blood was collected for cultures, clinically relevant rapid diagnostic tests when available, and dengue virus rapid diagnostic tests. Dengue virus infection remains a significant burden across Indonesia [28, 29], with disease incidence increasing in recent years [30]. Though other viral agents are present in Indonesia, none are as prevalent as dengue virus [24, 31], and most are challenging to diagnose due to limitations with available rapid diagnostic tests [32, 33]. Given the widespread prevalence of dengue virus infection, and the very high specificity (almost 100%) and good sensitivity (70–87%) of NS1 antigen rapid diagnostic tests [34], we employed universal dengue virus screening to rapidly resolve the unknown etiologies of study participants. Participants with negative NS1 antigen tests were further considered for BSIs through blood culture tests and other etiologies, as determined through advanced testing at the INA-RESPOND reference laboratory.

### Laboratory tests

Aerobic blood cultures were performed within 48 hours of a participant being admitted to the emergency department of a study site. Blood volumes of approximately 5–8 mL for adults and 1–3 mL for pediatrics were collected from each arm, whenever possible, directly into separate aerobic blood culture bottles. If blood could not be collected from each arm due to clinical reasons, blood was collected from a single arm for a single aerobic blood culture bottle. Study physicians were advised to delay the administration of IV antibiotics until blood specimens were collected, provided that there were no immediate risks to the participant. Each hospital performed a complete blood count (CBC) as part of standard-of-care procedures during enrollment.

Inoculated aerobic blood culture bottles were incubated using a continuous-monitoring blood culture system, either BACTEC (Becton-Dickinson, Sparks, Maryland) or BacT/Alert (bioMérieux, Inc., Durham, North Carolina) [35]. Manufacturer guidelines were followed for all bacterial cultures, and automated growth identification systems, either BD Phoenix (Becton Dickinson) or VITEK 2 (bioMérieux, Inc., Durham, North Carolina), were used for bacterial identification and antibiotic susceptibility testing. Blood cultures were performed and analyzed at the hospitals' nationally accredited clinical laboratories by trained, certified staff. All instruments and standards were calibrated appropriately according to manufacturer guidelines, and all tests were run alongside appropriate positive and negative control to ensure the integrity and accuracy of the results. Organism identification was considered acceptable when the confidence level in the automated growth identification system was ≥95% probability [36]. Quality control tests were performed weekly at all site laboratories, and each new lot of ID cards was tested using validated stocks of culture organisms.

Growth observed in blood cultures was classified as either "true positive" or "false positive." True positives included pathogenic bacterial species, particularly those identified as priority pathogens by the World Health Organization Global Antimicrobial Resistance and Use Surveillance System (WHO GLASS) [37], observed in at least one blood culture. Additionally, non-WHO GLASS pathogens found in either one or both cultures and being consistent with clinical manifestations were also considered to be true positives. False positives included growth of bacteria and fungi which were not clinically relevant and growth of known culture contaminants. Bacterial culture contamination was defined as any culture growing viridans group streptococci, *Corynebacterium spp.*, *Bacillus spp.*, *Diphtheroid spp.*, *Micrococcus spp.*, *Propionibacterium spp.*, and coagulase-negative staphylococci [12].

At the INA-RESPOND reference laboratory, specimens from all participants were screened for dengue using NS1 antigen ELISA, dengue RT-PCR, and dengue IgM and IgG. Molecular tests in acute specimens and serological tests in acute and convalescent specimens were performed to detect bacterial infections such as *S.* Typhi, *S.* Paratyphi A, *Leptospira spp.*, and *Rickettsia typhi*, and viruses such as influenza, chikungunya, and measles. Details of diagnostic assays for this study were previously described [19].

## Statistical analysis

Data were collected in OpenClinica (OpenClinica LLC, MA, USA) and analyzed using STATA v.15.1 (StataCorp LLC, TX, USA). Proportions were compared between categorical variables using Pearson's chi-squared test. The student's t-test was used to assess continuous variables. All p-values were two-sided with a significance level set to $p < 0.05$.

## Ethical clearance

Ethical approvals for the AFIRE study were granted by the Institutional Review Boards of the National Institute of Health Research and Development (NIHRD), Indonesia Ministry of Health (KE.01.05/EC/407/2012, dated 23 May 2012), the Faculty of Medicine at the University of Indonesia and RSUPN Dr. Cipto Mangunkusumo Hospital (451/PT02.FK/ETIK/2012, dated 23 July 2012), and RSUD Dr. Soetomo Hospital (192/Panke.KKE/VIII/2012, dated 13 August 2012). All eligible patients who agreed to participate in the study provided written informed consent before enrollment.

## Results

A total of 1,464 participants were enrolled in the AFIRE study, and aerobic blood cultures were performed for 1,459 participants (Fig 1). The remaining 5 participants had insufficient

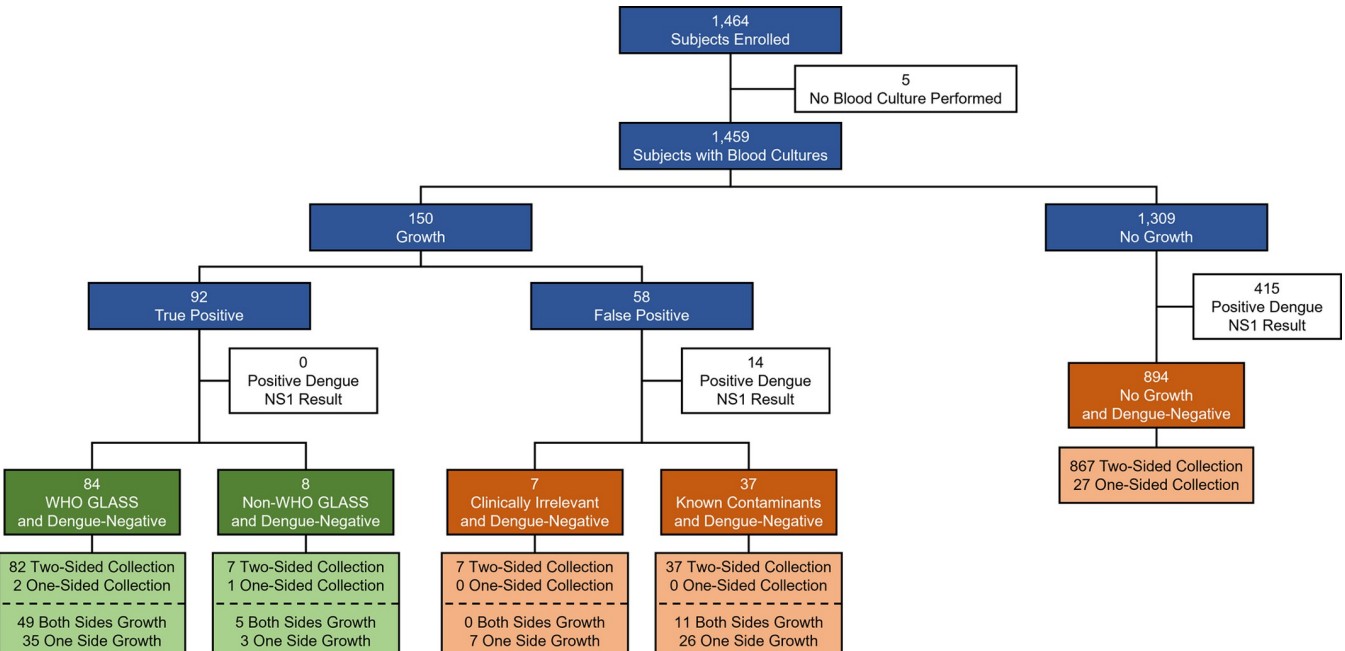

**Fig 1. General blood culture results observed among study participants.** Participants provided blood from either one or both arms for aerobic blood cultures, and bacterial growth was observed from either one or both sides. All participants providing blood underwent screening for dengue virus infection by NS1 antigen test.

blood specimens for following reasons: 1 adult was in a severe condition (decreased of consciousness), 2 participants (1 child and 1 adult) self-discharged against medical advice, and the guardians of 2 children refused to allow more blood to be drawn. Bacterial growth was observed for 10.3% (150) participants, including 56.0% (84) with WHO GLASS pathogens, 5.3% (8) with other non-WHO GLASS pathogens, and 38.7% (58) with false positives. No growth was observed for 89.7% (1,309) participants. All participants were screened for dengue virus by NS1 antigen and dengue IgM/IgG antibody tests, resulting in 29.4% (429) positive results, 415 from "No Growth" and 14 from the "False Positive" group. The remaining 70.6% (1,030) dengue-negative participants were included in this analysis.

## Results of blood cultures: Community-Acquired Infection (CAI)

Bacteremia was observed in 8.9% (92) of the 1,030 dengue-negative participants, with the most frequent pathogens being *S.* Typhi in 41 and *S.* Paratyphi A in 10 participants, *Escherichia coli* in 14 participants, and *Staphylococcus aureus* in 10 participants (Table 1). Dengue-negative false positive results were observed in 4.3% (44) participants, with the most frequent microorganism being contaminating coagulase-negative *Staphylococcus spp.* in 32 participants. From the 136 dengue-negative participants with any microbial growth, 97.8% (133) had blood collected from two sides of the body (Fig 1). Growth from both sides was observed in 58.7% of participants with true positive results and 25.0% of participants with false positive results.

Since *S.* Typhi and *S.* Paratyphi A were found in over half (55.4%) of true positives (Table 1), participants with true positive results were analyzed in either *Salmonella spp.* or non-*Salmonella spp.* groups (Table 2). Participant demographics revealed nearly equal numbers of male and female participants in the study, with equal numbers of true positive cases in the two groups. Participants in the *Salmonella spp.* group were significantly younger, with a median age of 14 years old, compared to non-*Salmonella spp.* and false positive groups, with

**Table 1. Specific blood culture results among dengue-negative study participants.**

| | Pathogen | Positive Results | Percent of Positive Results Within Group |
|---|---|---|---|
| WHO GLASS Priority Pathogens (N = 84) | *Salmonella spp.*<br>• *Salmonella enterica serovar Typhi* (41)<br>• *S. Paratyphi A* (10) | 51 | 60.7 |
| | *Escherichia coli* | 14 | 16.7 |
| | *Staphylococcus aureus* | 10 | 11.9 |
| | *Klebsiella pneumoniae* | 5 | 6.0 |
| | *Acinetobacter spp.* | 2 | 2.4 |
| | *Streptococcus pneumoniae* | 2 | 2.4 |
| Non-WHO GLASS Pathogens (N = 8) | *Pseudomonas aeruginosa* | 2 | 25.0 |
| | *Staphylococcus hominis ssp. hominis* | 1 | 12.5 |
| | *Enterobacter aerogenes* | 1 | 12.5 |
| | *Enterococcus faecalis* | 1 | 12.5 |
| | *Pseudomonas cepacea* | 1 | 12.5 |
| | *Pseudomonas spp.* | 1 | 12.5 |
| | *Streptococcus pyogenes* | 1 | 12.5 |
| Clinically Irrelevant Growth (N = 7) | *Pantoea spp.* | 2 | 28.6 |
| | *Sphingomonas paucimobilis* | 2 | 28.6 |
| | *Alcaligenes faecalis* | 1 | 14.3 |
| | *Candida pelliculosa* | 1 | 14.3 |
| | *Rhizobium radiobacter* | 1 | 14.3 |
| Contaminants (N = 37) | *Coagulase-Negative Staphylococcus* | 32 | 86.5 |
| | *Bacillus spp.* | 2 | 5.4 |
| | *Micrococcus luteus* | 1 | 2.7 |
| | *Kocuria spp.* | 1 | 2.7 |
| | *Streptococcus viridans* | 1 | 2.7 |
| No Growth (N = 894) | *None* | 0 | 0.0 |

median ages of 44 years old and 24.6 years old, respectively. Over 62.7% of *Salmonella spp.* cases were in participants ≤18 years old, while only 26.8% of non-*Salmonella spp.* cases were in this same age range. There were no significant differences between all groups in the days of onset before hospitalization or the length of hospitalization.

Intravenous antibiotics were administered prior to blood collection significantly less frequently in the *Salmonella spp.* group (17.6%, 9/51) compared to other groups (Table 2). All participants with true positive results were administered antibiotics following blood collection, and 74% of participants with false positive results received antibiotics. Hematology profiles at enrollment differed significantly between the *Salmonella spp.* and non-*Salmonella spp.* groups. Leukopenia and normal leukocyte counts were observed in 94.1% (48) of *Salmonella spp.* cases compared to 58.5% (24) of non-*Salmonella spp.* cases and 62.0% (582) of false positive and no growth cases. Similarly, leukocytosis was significantly lower in the *Salmonella spp.* group compared to the other groups. Lymphopenia was observed in 36.4% (16) of the *Salmonella spp.* cases, which is significantly lower than the 68.4% (26) non-*Salmonella spp.* cases and the 54.6% (442) false positive and no growth cases. Mortality was significantly higher in the non-*Salmonella spp.* group compared to the other groups.

Cases of true positives were distributed across age groups and study sites (Table 3). While *Salmonella spp.* were most frequently found in pediatrics (62.7% of cases), *E. coli*, *S. aureus*, and *K. pneumoniae* were most frequently found in adults (85.7%, 80.0%, and 80.0% of cases,

**Table 2. Participant characteristics, hematology results, and mortality.**

| | True Positive (92) | | False Positive and No Growth (938) | Total (1,030) |
|---|---|---|---|---|
| | *Salmonella spp.* (51) | Non-*Salmonella spp.* (41) | | |
| Male, N (%) | 29 (56.9) | 17 (41.5) | 502 (53.9) | 553 (53.7) |
| Median age, years (range, IQR) | 14 (2.5–54, 14.7) | 44 (1–84, 40.0) | 24.6 (1–92, 36.5) | 24 (1–92, 36.2) |
| Mean age, years (SD) | 16.2 (11.1)[D,E] | 39.6 (24.0)[D,F] | 28.6 (21.4)[E,F] | 28.5 (21.4) |
| Distribution of cases by age group, N (%) | | | | |
| 1–5 years | 4 (7.8) | 5 (12.2) | 154 (16.4) | 163 (15.8) |
| >5–18 years | 28 (54.9)[D,E] | 6 (14.6)[D] | 184 (19.6)[E] | 218 (21.2) |
| >18–45 years | 18 (35.3) | 11 (26.8) | 365 (38.9) | 394 (38.3) |
| >45–65 years | 1 (2.0)[D,E] | 13 (31.7)[C,D] | 179 (19.1)[C,E] | 193 (18.7) |
| >65 years | 0 (0.0)[B,D] | 6 (14.6)[C,D] | 56 (6.0)[B,C] | 62 (6.0) |
| Days of onset before hospitalization, median (range, IQR) | 7 (1–13, 4) | 4 (1–15, 4) | 4 (1–15, 4) | 4 (1–15, 4) |
| Length of hospitalization, median (range, IQR) | 7 (2–38, 4) | 8 (2–40, 7) | 6 (1–55, 3.3) | 6 (1–55, 4) |
| Received intravenous antibiotics prior to blood collection, N (%) | 9 (17.6)[A,E] | 16 (39.0)[A] | 389 (41.5)[E] | 414 (40.2) |
| Received any antibiotics following blood collection, N (%) | 31/31 (100)[E] | 18/18 (100)[A,C] | 199/269 (74.0)[A,C,E] | 248/318 (77.9) |
| Hematology at enrollment, N (%) | | | | |
| Leukopenia | 13/51 (25.5)[E] | 5/41 (12.2) | 120/937 (12.8)[E] | 138/1029 (13.4) |
| Normal Leukocyte | 35/51 (68.6)[A,E] | 19/41 (46.3)[A] | 462/937 (49.3)[E] | 516/1029 (50.1) |
| Leukocytosis | 3/51 (5.9)[D,E] | 17/41 (41.5)[D] | 355/937 (37.9)[E] | 375/1029 (36.4) |
| Lymphopenia | 16/44 (36.4)[B,D] | 26/38 (68.4)[D] | 442/810 (54.6)[B] | 484/892 (54.3) |
| Normal Lymphocyte | 17/44 (38.6)[A] | 7/38 (18.4)[A,C] | 285/810 (35.2)[C] | 309/892 (34.6) |
| Lymphocytosis | 11/44 (25.0)[E] | 5/38 (13.2) | 83/810 (10.2)[E] | 99/892 (11.1) |
| Outcome, N (%) | | | | |
| Died | 3 (5.9)[D] | 11 (26.8)[D,F] | 69 (7.4)[F] | 83 (8.1) |

Study participants with true positive culture results were sub-categorized into *Salmonella spp.* (consisted of *Salmonella enterica serovar* Typhi and Paratyphi A) and non-*Salmonella spp.* groups to better resolve analyses. Comparisons for significance occur across column groups only.

A,B,C indicates p-value <0.05

D,E,F indicates p-value <0.01

respectively). Most *Salmonella spp.* cases were seen in Bandung (BDG, 41.2%), Semarang (SMG, 23.5%), and Surabaya (SUB, 21.6%). This differed significantly from cases seen in Makassar (MKS, 9.8%), Yogyakarta (YOG, 2.0%), Denpasar (DPS, 2.0%), and Jakarta (JKT, 0.0%). Other than *Salmonella spp.*, there were no significant differences in the distribution of pathogens across study sites, likely due to the low numbers of cases.

The 938 participants in the false positive and no growth groups had specimens screened by other laboratory methods to determine potential etiologies (Table 4). PCR on blood specimens identified etiologies in 168 participants, serology identified etiologies in 220 participants, and other methods identified etiologies in 94 participants. Among the culturable bacterial pathogens identified in these groups were the WHO GLASS pathogens *S.* Typhi (41) and *S.* Paratyphi A (10), *S. pneumoniae* (18), *K. pneumonii* (8), *A. baumanii* (7), *E. coli* (7), and *S. aureus* (3). When combined with the culture results from the WHO GLASS priority pathogens group in Table 1, 50% of *S.* Typhi and *S.* Paratyphi A cases, 33.3% of *E. coli* cases, 23.1% of *S. aureus*

**Table 3. Positive blood culture pathogens by participant age group and study location.**

| Pathogen Identified | Age group (years old) | | | | | Location | | | | | | | Total |
|---|---|---|---|---|---|---|---|---|---|---|---|---|---|
| | ≥1–5 | >5–18 | >18–45 | >45–65 | >65 | Bdg | Sby | Smr | Dps | Mks | Yog | Jkt | |
| *Salmonella spp.* | 4 | 28 (1[†]) | 18 (1[†]) | 1 (1[†]) | 0 | 21 | 11 | 12 | 1 | 5 | 1 | 0 | 51 |
| *Escherichia coli* | 1 | 1 | 3 (1[†]) | 5 (1[†]) | 4 | 3 | 3 | 0 | 4 | 0 | 3 | 1 | 14 |
| *Staphylococcus aureus* | 0 | 2 | 4 (1[†]) | 4 (1[†]) | 0 | 1 | 1 | 3 | 2 | 1 | 0 | 2 | 10 |
| *Klebsiella pneumoniae* | 0 | 1 | 0 | 3 (2[†]) | 1 | 0 | 1 | 0 | 2 | 2 | 0 | 0 | 5 |
| *Acinetobacter spp.* | 0 | 1 | 1 | 0 | 0 | 0 | 0 | 0 | 1 | 0 | 0 | 1 | 2 |
| *Enterobacter aerogenes* | 0 | 0 | 1 (1[†]) | 0 | 0 | 0 | 0 | 0 | 1 | 0 | 0 | 0 | 1 |
| *Enterococcus faecalis* | 1 | 0 | 0 | 0 | 0 | 0 | 0 | 0 | 0 | 0 | 0 | 1 | 1 |
| *Pseudomonas aeruginosa* | 1 | 0 | 1 (1[†]) | 0 | 0 | 0 | 0 | 0 | 0 | 0 | 2 | 0 | 2 |
| *Pseudomonas cepacea* | 0 | 0 | 0 | 0 | 1 | 1 | 0 | 0 | 0 | 0 | 0 | 0 | 1 |
| *Pseudomonas species* | 0 | 0 | 1 | 0 | 0 | 0 | 1 | 0 | 0 | 0 | 0 | 0 | 1 |
| *Streptococcus pneumoniae* | 1 (1[†]) | 1 (1[†]) | 0 | 0 | 0 | 1 | 0 | 0 | 0 | 1 | 0 | 0 | 2 |
| *Streptococcus pyogenes* | 0 | 0 | 0 | 1 | 0 | 0 | 0 | 0 | 1 | 0 | 0 | 0 | 1 |
| *Staphylococcus hominis ssp hominis* | 1 (1[†]) | 0 | 0 | 0 | 0 | 0 | 0 | 0 | 0 | 0 | 1 | 0 | 1 |
| **Total** | 9 (2[†]) | 34 (2[†]) | 29 (5[†]) | 14 (5[†]) | 6 | 27 | 17 | 15 | 12 | 9 | 7 | 5 | 92 |

[†] Indicates study participants who died

Bdg: Bandung; Sby: Surabaya; Smr: Semarang; Dps: Denpasar; Mks: Makassar; Yog: Yogyakarta; Jkt: Jakarta

cases, 61.5% of *K. pneumoniae* cases, 77.8% of *Acinetobacter spp.* cases, and 90% of *S. pneumoniae* cases in the AFIRE study [19] were not identified by blood cultures.

## Antimicrobial resistance patterns

Antimicrobial resistance patterns were observed in several participants with blood cultures positive for WHO GLASS priority pathogens (Fig 2). Among the 10 *S.* Paratyphi A cases, evidence of multidrug resistance was observed in one participant and monoresistance in one participant. In contrast, *E. coli* cases were mostly multidrug resistant (42.9%, 6/14) or monoresistant (14.3%, 2/14), with observed resistances to ampicillin (87.5%, 7/8), co-trimoxazole (60.0%, 3/5), ceftriaxone (45.4%, 5/11), ceftazidime (41.6%, 5/12), cefotaxime (37.5%, 3/8), cefepime (33.3%, 2/6), ciprofloxacin (30.0%, 3/10), and levofloxacin (25.0%, 2/8). Two participants (JOG-A and DPS-A) receiving ceftriaxone died before their antimicrobial resistance test results, and one participant (JOG-B) survived when switched from ceftazidime to ciprofloxacin based on their test results.

Methicillin-resistant *S. aureus* (MRSA) was observed in one participant based on oxacillin susceptibility testing, and two participants with oxacillin-sensitive *S. aureus* infections died. Both participants with *S. pneumoniae* bacteremia died, though antimicrobial resistance was only observed in one of the participants. All cases of *Acinetobacter spp.* and *K. pneumoniae* that underwent drug sensitivity testing were sensitive to antibiotics.

## Disease outcomes

Characteristics and laboratory findings of participants who died during hospitalization are shown in Table 5. A total of 83 participants in this analysis died during hospitalization. Among these, 16.9% (14) had true positive blood cultures (Table 5A), resulting in 15.2% mortality in the true positive group. This mortality rate is twofold higher than the 7.4% mortality observed in the false positive and no growth groups. Overall mortality in the *S. Typhi.* group (5.9%) was significantly lower than the non-*Salmonella spp.* group (26.8%). Among deceased participants,

**Table 4. Pathogens detected by molecular, serological, or other laboratory methods from participants with false positive and no growth blood cultures.**

| False Positive and No Growth (N = 938) | | Confirmatory Methods | | |
|---|---|---|---|---|
| Pathogen | N | Blood PCR | Serology | Other Methods |
| *Rickettsia typhi* | 101 | 65 | 36 | |
| Influenza | 66 | 0 | 59 | 7: Sputum PCR |
| *Salmonella enterica serovar* Typhi and Paratyphi A | 51 | 3 | 48 | |
| *Leptospira spp.* | 44 | 31 | 13 | |
| Chikungunya | 38 | 30 | 8 | |
| Dengue | 35 | 0 | 35 | |
| *Mycobacterium tuberculosis* | 20 | 0 | 0 | 20: Sputum Microscopy |
| *Streptococcus pneumoniae* | 18 | 10 | 0 | 8: Sputum PCR |
| Measles | 14 | 9 | 5 | |
| Amoeba | 11 | 0 | 0 | 11: Feces Microscopy |
| RSV | 11 | 0 | 9 | 2: Swab PCR |
| HHV-6 | 9 | 9 | 0 | |
| *Klebsiella pneumoniae* | 8 | 1 | 0 | 5: Sputum Culture<br>2: Swab Culture |
| *Acinetobacter baumanii* | 7 | 1 | 0 | 4: Sputum PCR<br>1: Swab PCR<br>1: Urine PCR |
| *Escherichia coli* | 7 | 1 | 0 | 4: Urine Culture<br>2: Pus Culture |
| Hepatitis A | 6 | 0 | 6 | |
| *Pseudomonas aeruginosa* | 6 | 0 | 0 | 4: Sputum Culture<br>2: Urine Culture |
| *Enterococcus faecalis* | 3 | 0 | 0 | 2: Pus Culture<br>1: Urine Culture |
| *Staphylococcus aureus* | 3 | 0 | 0 | 3: Pus Culture |
| *Mycobacterium leprae* | 2 | 0 | 0 | 2: Skin Microscopy |
| *Plasmodium spp.* | 2 | 0 | 0 | 2: Rapid Antigen Test |
| Seoul virus | 2 | 2 | 0 | |
| Adenovirus | 1 | 1 | 0 | |
| *Ascaris lumbricoides* | 1 | 0 | 0 | 1: Feces Microscopy |
| *Ascaris lumbricoides* and *Trichuris Trichiura* | 1 | 0 | 0 | 1: Feces Microscopy |
| *Bordetella pertussis* and *Streptococcus pneumoniae* | 1 | 0 | 0 | 1: Sputum PCR |
| HCoV-OC43 | 1 | 1 | 0 | |
| *Enterobacter aerogenes* | 1 | 0 | 0 | 1: Sputum Culture |
| *Enterobacter cloacae* | 1 | 0 | 0 | 1: Sputum Culture and PCR |
| *Enterococcus avium* | 1 | 0 | 0 | 1: Pus Culture |
| Enterovirus | 1 | 1 | 0 | |
| EPEC | 1 | 0 | 0 | 1: Feces Culture |
| HIV | 1 | 1 | 0 | |
| Metapneumovirus | 1 | 0 | 0 | 1: Swab PCR |
| *Moraxella catarrhalis* and Influenza B | 1 | 0 | 0 | 1: Sputum Culture and PCR |
| *Mycoplasma pneumoniae* | 1 | 0 | 0 | 1: Sputum PCR |
| Norovirus II | 1 | 1 | 0 | |
| *Rickettsia felis* | 1 | 1 | 0 | |
| Rubella | 1 | 0 | 1 | |
| *Streptococcus faecalis* | 1 | 0 | 0 | 1: Urine Culture |
| Unknown | 456 | 0 | 0 | |

*(Continued)*

**Table 4.** (Continued)

| False Positive and No Growth (N = 938) | | Confirmatory Methods | | |
|---|---|---|---|---|
| Pathogen | N | Blood PCR | Serology | Other Methods |
| Total | 938 | 168 | 220 | 94 |

Plasma, serum, and clinically relevant specimens were collected from all study participants and tested in a central lab for culturable and non-culturable pathogens based on a standard study algorithm and clinical suspicion.

there were no significant differences in demographics between the true positive group and false positive and no growth groups. Most deceased participants had comorbidities including diabetes mellitus (4), hepatitis B (3), HIV (2), tuberculosis (2), brain tumor (1), TRALI (1), neoplasia (1), and others (6) (Table 5B). Antimicrobial-resistant pathogens were identified in 3 of the 14 deceased participants with true positives (Table 5). In the false positive and no growth groups, other laboratory methods such as PCR and/or serology were used to identify culturable bacterial pathogens including *S.* Typhi (2), *A. baumanii* (1), *E. avium* (1), *E. coli* (1), *M. catar-rhalis* (1), and *S. pneumoniae* (1) (Table 5B).

## Discussion

BSI causes a high burden of morbidity and mortality worldwide, particularly in low- and middle-income countries (LMICs). Exact figures for BSI incidence and associated mortality in LMICs are challenging to find due to the lack of bacteriological laboratories and routine surveillance systems [38, 39]. In Indonesia, very few acute febrile patients undergo aerobic blood culture testing since it is not standard practice in the healthcare system, largely due to resource and capacity restrictions [17]. The AFIRE study presents a unique opportunity to improve our understanding of BSIs in the country since aerobic blood cultures were performed on nearly all participants, regardless of clinical suspicion of bacteremia.

**Fig 2. Antimicrobial resistance patterns observed in WHO GLASS priority pathogens from true positive blood cultures.** Participants with resistant (R) infections are identified by study location, and participants with sensitive (S) infections or infections with no testing data (ND) are grouped into Other or No Data categories.

**Table 5. Participant characteristics, clinical diagnoses, and identified pathogens from fatal cases in the study.**

| (A) Characteristics of deceased participants categorized by blood culture growth result. | | | | |
|---|---|---|---|---|
| | **True Positive (14)** | | **False Positive and No Growth (69)** | **Total (83)** |
| | *Salmonella enterica serovar Typhi* (3) | **Non-*Salmonella spp.* (11)** | | |
| Male, N (%) | 3 (100) | 7 (63.6) | 36 (52.2) | 46 (55.4) |
| Distribution of cases by age group, N (%) | | | | |
| 1–5 years | 0 (0.0) | 2 (18.2) | 4 (5.8) | 6 (7.2) |
| >5–18 years | 1 (33.3) | 1 (9.1) | 7 (10.1) | 9 (10.8) |
| >18–45 years | 1 (33.3) | 4 (36.4) | 24 (34.8) | 29 (34.9) |
| >45–65 years | 1 (33.3) | 4 (36.4) | 25 (36.2) | 30 (36.1) |
| >65 years | 0 (0.0) | 0 (0.0) | 9 (13) | 9 (10.8) |
| Received intravenous antibiotics prior to blood collection, N (%) | 1 (33.3) | 1 (9.1) | 34 (49.3) | 36 (43.4) |
| Length of hospitalization, median (range, IQR) | 4 (2–38) | 12 (2–17) | 8 (2–54) | 8 (2–54) |
| Comorbidities, N (%) | 2 (66.6) | 10 (90.9) | 60 (86.9) | 72 (86.7) |

| (B) Pathogens from fatal cases confirmed by blood culture or other lab methods and the accompanying clinical diagnoses, participant comorbidities, and AMR observations | | | |
|---|---|---|---|
| **True Positive (14)** | **Clinical Diagnosis at Death** | **Comorbidities** | **Antimicrobial Resistance** |
| *Salmonella enterica serovar* Typhi (3) | Typhoid fever | Hepatitis B, HIV, TB | None |
| | Acute limb ischemia | Acute Limb Ischemia | None |
| | Sepsis, typhoid fever | Transfusion-Related Acute Lung Injury (TRALI) | None |
| *Escherichia coli* (2) | Cholangitis | Diabetes, Hepatitis B | Yes |
| | Sepsis | Anemia | Yes |
| *Klebsiella pneumoniae* (2) | UTI, diabetic ketoacidosis | Diabetes | None |
| | UTI | Stroke | None |
| *Staphylococcus aureus* (2) | UTI | Diabetes | None |
| | Sepsis | Diabetes, Chronic Kidney Disease | None |
| *Streptococcus pneumoniae* (2) | Aseptic meningitis, acute otitis media | Epilepsy | Yes |
| | | Myelodysplasia, Hepatitis B (Cirrhosis) | None |
| *Pseudomonas aeruginosa* (1) | Stevens-Johnson syndrome | HIV, TB, Toxoplasmosis | No data |
| *Enterobacter aerogenes* (1) | Cholangitis, Sepsis | None | No data |
| *Staphylococcus hominis ssp hominis* (1) | | Craniopharyngioma | None |
| **False Positive and No Growth (69) [Confirmatory Methods]** | **Clinical Diagnosis at Death** | | |
| *Mycobacterium tuberculosis* (8) [GeneXpert (2), Microscopy (6)] | Pulmonary TB (3), Colitis TB and Spondylitis TB, Millar TB, HIV, Community-acquired Pneumonia, Sepsis | | |
| *Rickettsia typhi* (6) [PCR (6)] | Sepsis (3), Community-acquired Pneumonia, Meningoencephalitis, Diabetic Neuropathy | | |
| Influenza (3) [PCR (2), Serology (1)] | Bronchiectasis, Community-acquired Pneumonia, Sepsis | | |
| *Salmonella* Typhi (2) [Serology (2)] | Hirschsprung's disease, HIV | | |
| *Acinetobacter baumanii* (1) [Sputum PCR] | Community-acquired Pneumonia | | |
| *Ascaris lumbricoides* (1) [Microscopy] | Typhoid Fever | | |
| *Enterococcus avium* (1) [Pus culture] | Diabetic Ulcer | | |
| *Escherichia coli* (1) [Urine culture] | UTI | | |
| HIV (1) [PCR] | Sepsis | | |
| *Leptospira spp.* (1) [PCR] | Dengue Hemorrhagic Fever I | | |
| *Moraxella catarrhalis* and Influenza B (1) [Sputum culture and sputum PCR] | Community-acquired Pneumonia | | |
| RSV (1) [Serology] | TB Pleuritis | | |

(*Continued*)

**Table 5.** (Continued)

| | |
|---|---|
| *Streptococcus pneumoniae* (1) [Sputum PCR] | Community-acquired Pneumonia |
| Unknown (41) [None] | HIV (6), Sepsis (6), Community-acquired Pneumonia (9), Cellulitis (2), Cholangitis (2), Lung Abscess, Acute Leukemia, Bacterial Meningitis, Bronchitis, Cholecystitis, Chronic Myelocytic Leukemia, COPD, Diarrhea, Extrapulmonary TB, GEA, Hepatitis B, Pancytopenia, SLE, Typhoid Fever, UTI, Unknown |

Microbial growth was observed in 10.3% of all participants, with bacteremia being ultimately confirmed in 6.3% of all participants (Fig 1). These proportions are similar to previous reports, where positivity rates ranged from 10.0–11.4% [17]. The high prevalence of dengue fever in Indonesia often complicates the clinical assessment of acute febrile illness [25], so specimens from all participants in the AFIRE study were retrospectively tested for dengue NS1 antigen to exclude dengue as a cause of illness [19]. Data on co-infections with dengue virus and bacteremia is limited. A literature review of published case reports and studies from January 1943 to March 2016 found 3 studies in Singapore and Taiwan reporting concurrent bacteremia in 0.18–7% of dengue fever cases [40–42]. A concurrent dengue virus and *S.* Typhi case was also reported from Bandung, Indonesia [43]. In all of these studies, blood was collected for bacterial culture because patients did not improve clinically a few days to a week after dengue fever was diagnosed. Furthermore, in the majority of cases, dengue virus infection was confirmed by serology only (IgM detected or four-fold IgG increase). These reports support our finding that simultaneous infection with bacteria and dengue virus is rare. In our study, bacterial growth observed in 14 participants with positive dengue NS1 antigen tests were considered false positive blood cultures (5 *Staphylococcus hominis*, 4 *Staphylococcus epidermidis*, 1 *Kocuria rosea*, 1 *Micrococcus aureus*, 1 *Staphylococcus arlettae*, 1 coagulase-negative *Staphylococcus spp*., and 1 *Staphylococcus waneri*).

Among dengue-negative participants with any microbial growth, 97.8% had blood cultures performed from two sides of collection. One-sided blood culture lacks sufficient sensitivity for BSI detection [44], and two-sided cultures make it easier to distinguish true bacteremia and contamination [44, 45]. It has been demonstrated that collecting two or more blood culture sets, each comprising two bottles, over twenty-four hours will detect over 94% of bacteremia episodes, compared to a detection rate of only 73% with the first blood culture [44]. In many developing countries, collecting multiple blood culture sets is generally not feasible, but the minimum practice of a single, one-sided blood culture still has value if clinical care teams understand its limitations. Our data suggest that, in situations where a single, one-sided blood culture is performed, the likelihood of missing a case of bacteremia is 39% (35/89) (8.9% (89/1000) vs 5.4% (54/1000) (Fig 1). Indonesian clinicians should consider this reduced sensitivity when acting on culture results.

The reliability and interpretation of blood culture results is significantly affected by both contamination rates and the use of antibiotics prior to blood collection. General target rates for culture contamination have been set at 3% [45], and in our study we observed an overall contamination rate of 3.6%. These findings are consistent with previous reports, including a 2010–2013 study at Sardjito Hospital in Yogyakarta that found a contamination rate of 4.1% in children at the pediatric ICU and in pediatric wards [46]. Additional reports from rural Thailand and Taiwan found contamination rates ranging from 4.1–6.1% and 2.6%, respectively [47, 48]. The proportion of participants who were given intravenous antibiotics prior to blood collection in our study was high (40.2%), and this may alter the blood culture results considerably [49, 50]. In Indonesia, antibiotic therapy is often initiated preemptively and without confirmatory testing in an attempt to maximize positive clinical outcomes [51]. This broad use of antibiotics likely masks the true prevalence of bacteremia and may have negative consequences

for patients who subsequently appear to have no infection. Among participants with false positives or no growth, 111 had culturable microbes confirmed by other methods (Table 4), 7 of which died (Table 5). 56.8% of these overall participants received antibiotics prior to blood collection. The expansion of molecular methods would significantly help to tackle this problem, as nucleic acid probe and amplification tests have been shown to significantly improve the speed and accuracy of results in blood stream infections even after antibiotic use [52, 53].

White blood cell counts, particularly leukopenia and leukocytosis, have been used to predict blood culture results. However, the accuracy of systemic inflammatory response syndrome (SIRS) criteria [54], Shapiro criteria [55], and the quick Sequential Organ Failure Assessment (qSOFA) score [56] could not be confirmed in our study. This is primarily due to the significant difference in leukocyte profiles between participants with *Salmonella spp.* versus non-*Salmonella spp.* infections. Our study suggests, as proposed by Ombelet [57] and Seigel [58] that leukocytosis should not be used as a predictor for positive blood cultures in *S. enterica*-endemic areas.

We found that *S.* Typhi and *S.* Paratyphi A infection was the most common community-acquired BSI (Table 1) at 55.4% of cases, which aligns with previous studies conducted in limited-resource environments [46, 47]. The majority of *S.* Typhi and *S.* Paratyphi A bacteremia was in pediatrics, which is consistent with a previous report from a blood culture study in Jakarta where the incidence rate of typhoid fever was higher in the 2–15 year age group, with a mean age of onset of 10.2 years [59]. This commonly observed age association may be due to poor hygiene practices or the consumption of foods, particularly street food, outside of the home [60]. Though over half of bacteremia cases were due to *S.* Typhi and *S.* Paratyphi A infection, only 21.4% of bacteremia deaths were due to the pathogen. Among these fatal cases, all had significant comorbidities, suggesting that patients with multiple comorbidities would benefit from prioritization of blood culture diagnostics.

Despite the high prevalence of *S.* Typhi and *S.* Paratyphi A among participants with bacteremia, previous reports have found the overall sensitivity of blood cultures to be only 66% (95% CI 56–75%) when compared to more sensitive tests such as bone marrow cultures [61]. Though bone marrow cultures were not performed as part of our study, further molecular and serological testing as part of the AFIRE study identified an additional 51 cases in the false positive and no growth groups (Table 4), 2 of which were fatal. Most participants with negative blood cultures and false positive results (41.5%) had already received IV antibiotics prior to blood collection, which may have substantially diminished the yield of blood cultures [49, 50]. While blood collection prior to antibiotic administration is ideal, an environment like Indonesia, where preemptive antibiotic use is common, would significantly benefit from supplementing blood culture testing with molecular and serological tests. These tests do have drawbacks, as molecular diagnostics can have poor sensitivity due to the low organism burden in bodily fluids [62], and serological diagnostics require increasing titers in convalescent specimens compared to acute specimens given high background antibody levels in endemic regions [63]. Further research on combining a clinical prediction algorithm with disease-specific blood cultures for patients with febrile illnesses in typhoid-endemic areas could be a potential route to improve patient outcomes in a community-based setting while waiting for the wider adoption of molecular and serological testing. Among cases of *S.* Typhi and *S.* Paratyphi A bacteremia, the prevalence of antimicrobial resistance to the antibiotic of choice was only 3.9% (Fig 2), which is similar to previous studies in Indonesia [64–66]. In the 2011–2015 period, rates of resistance against most antimicrobials for *S.* Typhi and *S.* Paratyphi A were low, indicating that there is a distinct epidemiological dynamic of enteric fever in Indonesia compared to the rest of the world [64, 67]. This could be due to different strains of *S.* Typhi and *S.* Paratyphi A

which may possess different genes that contribute to resistance [64, 65], though we did not perform genotyping or sequencing as part of our study.

In addition to *S.* Typhi and *S.* Paratyphi A bacteremia, we identified cases of bacteremia caused by other WHO GLASS and non-GLASS pathogens. *E. coli* was the second most common cause of BSI, with over half of isolates possessing some form of antimicrobial resistance. Both fatal cases were found to possess third-generation cephalosporin (3GC) and fluoroquinolone resistance. The global incidence of community-acquired BSI due to *E. coli* is relatively high, with an estimated 50–60 cases per 100,000 population [68–70], and the proportion of 3GC resistance has reached levels >60% in some parts of the world [71, 72]. We found 3GC-resistance rates of 35.7% in our study, which is consistent with the WHO GLASS report of 36.6% (interquartile range [IQR] 17.5–58.3) [37]. The fluoroquinolone-resistance rates of 22% that we observed were high but consistent with previous reports from Indonesia [73, 74].

Bacteremia from *S. aureus* infection was found in 10.9% cases in our study, and the observed mortality rate of 20% was consistent with a previous report [75]. Both participants who died were diabetic and contracted oxacillin-sensitive infections, suggesting that the cause of death may have been due more to the timing of diagnosis and treatment. It is well-known that diabetics are at high risk for infections with *S. aureus* [76], so comorbidities should be strongly considered when prioritizing blood culture testing. Two participants with systemic lupus erythematosus (SLE) developed *S. aureus* BSIs, which has been associated with classic hyper-IgE syndrome [77]. The colonization of *S. aureus* in the body often increases in patients with SLE and may predispose them to BSI, worsening the SLE itself and leading to a feedback loop with the potential to reinforce autoimmune symptoms [78, 79]. The proportion of MRSA in our study (10%) was lower than the WHO GLASS report (24.9% (IQR 11.4–42.7)) [37], though this is understandable given that our study was not a systematic surveillance of *S. aureus* infections across the country. Geographic variation of CAI with MRSA has been observed in the Asia-Pacific region, including Taiwan, the Philippines, Vietnam, and Sri Lanka (30–39%); Korea and Japan (15–20%); and Thailand, India, and Hong Kong (3–9%) [80, 81]. Data from Indonesia remains limited, but a recent study has shown that the carriage rate of MRSA in the nose and throat of patients admitted to surgery and internal medical wards at Dr. Soetomo Hospital in Surabaya was 8.1% among 643 patients [82]. Additionally, a report on 259 *S. aureus* isolates collected from clinical cultures of patients at four tertiary care hospitals in Denpasar, Malang, Padang, and Semarang found that 6.6% and 18.5% were MRSA and PVL-positive methicillin-susceptible *S. aureus*, respectively [83].

Besides *E. coli* and *S. aureus*, we observed the other WHO GLASS pathogens *K. pneumonia*, *S. pneumonia*, and *Acinetobacter spp*. in our study. *K. pneumonia* was mostly found in patients with UTI and respiratory illnesses. The two fatal cases were most likely associated with the participants' chronic illnesses (stroke and kidney failure), as none of the isolates were 3GC, fluoroquinolone, or co-trimoxazole resistant. Both cases of *S. pneumonia* bacteremia were found in pediatric participants, and both were fatal. The participant with a penicillin-sensitive infection had myelodysplasia syndrome, and the participant with a ceftriaxone-resistant infection had clinical meningitis. *S. pneumonia* was also found by molecular methods in 8 participants whose blood cultures were negative, supporting a previous report that successful diagnostic approaches using blood cultures alone are difficult because of reduced sensitivity [84]. *Acinetobacter lwoffii* was identified in two participants, both having gastro-intestinal symptoms and receiving an initial diagnosis of typhoid fever. Treatment with cefixime resolved the infections. A similar case with fever, abdominal pain, and diarrhea has been reported in a 64 year-old man in Texas, USA [85].

Our study found the most frequent BSI pathogens to be *S.* Typhi and *E. coli*, though multi-drug-resistant *E. coli* was the most problematic. The challenges of AMR in Indonesia are

similar to those of many other low and middle-income countries in the region and globally [20]. Misuse and overuse of antibiotics in humans, livestock, and aquaculture may be the key drivers of resistance in the country [86]. Despite current policies related to antimicrobial use in Indonesia, frequent and unnecessary prescription of antibiotics by physicians, high rates of self-medication, and over-the-counter access to antibiotics remain common [87]. Since 2016, the Indonesia Ministry of Health has boosted their AMR stewardship program to tackle this growing challenge, directing substantial funding to the national AMR control committee [20]. Further support for AMR prevention and the alignment of national policies with global policies and standards will substantially improve the growing challenge of AMR infections in Indonesia.

Our study has several limitations. First, the blood specimens analyzed as part of this study were collected only from a limited number of extremely ill patients admitted to tertiary hospitals. Blood culture positivity rates, AMR patterns, and clinical outcomes may not be generalizable to the Indonesian population at-large, though better understanding this critically ill population will hopefully lead to a reduction in mortality from BSIs. Second, only aerobic blood cultures were performed, which may have resulted in missed BSIs caused by anaerobic bacterial. The generally low yield of anaerobic bacteria combined with increasing costs and volumes of blood drawn [13, 88, 89] make anaerobic cultures impractical for many hospitals in Indonesia. In the future, rationally targeting the use of anaerobic culture bottles based on careful clinical assessment may result in substantial savings and facilitate the broader adoption of the diagnostic in the country [90]. Lastly, AMR susceptibility testing in this study was performed and reported according to general practice in Indonesia, as our study was not initially designed as an AMR study. Consequently, our data has substantial gaps and missing information. A standardized approach and electronic results reporting system in Indonesia would significantly improve the accuracy and utility of AMR susceptibility testing.

## Conclusion

We presented aerobic blood culture findings from a multi-centre study of patients with acute febrile illness admitted to eight major hospitals across Indonesia. Our universal use of aerobic blood cultures is unique in Indonesia, the results of which help clarify the epidemiology and burden of BSI, rates of contamination among CAI, and common AMR patterns in Indonesia. Bacteremia was observed in 8.9% participants, with the most frequent pathogens being *S*. Typhi and *S*. Paratyphi A, *E. coli*, and *S. aureus*. Two *S*. Paratyphi A cases had evidence of AMR, and several *E. coli* cases were multidrug resistant (42.9%) or monoresistant (14.3%). Culture contamination was observed in 3.6% cases. Our data suggest that blood cultures should be included as a routine diagnostic test, and pre-screening patients for the most common viral infections, such as dengue, influenza and chikungunya viruses, would conserve scarce resources without negatively impacting patient benefit. The routine practice of AMR susceptibility testing on positive blood cultures in Indonesia is encouraging and should be continued to inform clinical decisions on patient treatment in real-time. The country could benefit from clear guidance at the national level, particularly regarding the timing of blood collection prior to antibiotic administration, the prioritization of patients with comorbidities, blood collection practices to reduce environmental contamination, and the supplementation of blood cultures with molecular assays to combat false-negative results. Additionally, Indonesia could greatly benefit from a nationwide program for the systematic collection and dissemination of blood culture and AMR results.

## Supporting information

**S1 Dataset.**
(XLSX)

## Acknowledgments

We would like to thank all of the patients who participated in this study, the site study teams and investigators, US-NIAID and Indonesia NIHRD, the Indonesian Ministry of Health, the INA-RESPOND Network Steering Committee, and the sample repository team.

## Author Contributions

**Conceptualization:** Pratiwi Soedarmono, Aly Diana, Patricia Tauran, Dewi Lokida, Abu Tholib Aman, Bachti Alisjahbana, Dona Arlinda, Emiliana Tjitra, Herman Kosasih, Ketut Tuti Parwati Merati, Mansyur Arif, Muhammad Hussein Gasem, Nurhayati Lukman, Retna Indah Sugiyono, Usman Hadi, Vivi Lisdawati, Muhammad Karyana.

**Data curation:** Pratiwi Soedarmono, Patricia Tauran, Dewi Lokida, Abu Tholib Aman, Bachti Alisjahbana, Ketut Tuti Parwati Merati, Mansyur Arif, Muhammad Hussein Gasem, Usman Hadi, Vivi Lisdawati.

**Formal analysis:** Pratiwi Soedarmono, Aly Diana, Patricia Tauran, Dewi Lokida, Dona Arlinda, Emiliana Tjitra, Herman Kosasih, Nugroho Harry Susanto, Nurhayati Lukman, Karine G. Fouth Tchos.

**Investigation:** Patricia Tauran, Abu Tholib Aman, Ketut Tuti Parwati Merati, Mansyur Arif, Vivi Lisdawati.

**Methodology:** Pratiwi Soedarmono, Patricia Tauran, Dewi Lokida, Abu Tholib Aman, Bachti Alisjahbana, Dona Arlinda, Herman Kosasih, Ketut Tuti Parwati Merati, Mansyur Arif, Muhammad Hussein Gasem, Nugroho Harry Susanto, Retna Indah Sugiyono, Usman Hadi, Vivi Lisdawati, Aaron Neal, Muhammad Karyana.

**Project administration:** Muhammad Karyana.

**Resources:** Dewi Lokida, Bachti Alisjahbana, Emiliana Tjitra, Usman Hadi, Karine G. Fouth Tchos, Aaron Neal.

**Supervision:** Pratiwi Soedarmono, Dewi Lokida, Bachti Alisjahbana, Emiliana Tjitra, Herman Kosasih, Karine G. Fouth Tchos, Aaron Neal.

**Validation:** Pratiwi Soedarmono, Herman Kosasih, Aaron Neal.

**Visualization:** Aly Diana, Nurhayati Lukman, Aaron Neal.

**Writing – original draft:** Aly Diana, Patricia Tauran, Herman Kosasih, Nugroho Harry Susanto, Nurhayati Lukman, Karine G. Fouth Tchos, Aaron Neal.

**Writing – review & editing:** Pratiwi Soedarmono, Aly Diana, Patricia Tauran, Dewi Lokida, Abu Tholib Aman, Bachti Alisjahbana, Dona Arlinda, Emiliana Tjitra, Herman Kosasih, Ketut Tuti Parwati Merati, Mansyur Arif, Muhammad Hussein Gasem, Nugroho Harry Susanto, Nurhayati Lukman, Retna Indah Sugiyono, Usman Hadi, Vivi Lisdawati, Karine G. Fouth Tchos, Aaron Neal, Muhammad Karyana.

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
