## [Decision Letter · Decision Letter 0]

30 May 2022

PONE-D-22-07517The Characteristics of Bacteremia among Patients with Acute Febrile Illness Requiring Hospitalization in IndonesiaPLOS ONE

Dear Dr. Kosasih,

Thank you for submitting your manuscript to PLOS ONE. After careful consideration, we feel that it has merit but does not fully meet PLOS ONE’s publication criteria as it currently stands. Therefore, we invite you to submit a revised version of the manuscript that addresses the points raised during the review process.

We look forward to receiving your revised manuscript.

Kind regards,

Dwij Raj Bhatta, PhD

Academic Editor

PLOS ONE

Journal Requirements:

Additional Editor Comments (if provided):

Submitted manuscript on very important study requires corrections in each section as pointed out by reviewers! Specially adress all the comments made by reviewers& answer important questions raised!

Reviewers' comments:

Reviewer's Responses to Questions

**Comments to the Author**

1. Is the manuscript technically sound, and do the data support the conclusions?

Reviewer #1: Yes

Reviewer #2: Yes

2. Has the statistical analysis been performed appropriately and rigorously? 

Reviewer #1: Yes

Reviewer #2: Yes

3. Have the authors made all data underlying the findings in their manuscript fully available?

Reviewer #1: Yes

Reviewer #2: Yes

4. Is the manuscript presented in an intelligible fashion and written in standard English?

Reviewer #1: Yes

Reviewer #2: No

5. Review Comments to the Author

Reviewer #1: This compilation of data from different centers over many years is commendable. This highlights the issues faced in diagnostic microbiology in developing countries. It is an interesting paper with important observations and discussions. Some spellings need review and correction. Recommend to submit after corrections.

Reviewer #2: The Characteristics of Bacteremia among Patients with Acute Febrile Illness Requiring

Hospitalization in Indonesia.

Evaluation. This report addresses an important subject in Bacteriemia and Acute Febrile illness; i.e., the worrying trend of antimicrobial resistance in bacterial pathogens (Salmonella and Non Salmonella spp) . It reports the frequency and distribution of bacterial pathogens in blood culture and its susceptibility pattern isolated from various specimens from a seven medical center in Indonesia, from which similar reports are scarce. Though it is better attempt by Soedarmono et al., to know information on bacteremia and other causative agent of Acute Febrile illeness in Indonesia.

Comments

1.Give rationale of the study? Why is NS1 antigen screening only performed? What about other viral agents related AFI?

2.Why you performed Blood culture 0f 1459 Cases? You have mentioned 1464 were enrolled? What about 5??

3.At the end of introduction, please give some update of Acute Febrile illness and their epidemiology in Indonesia.

4.Which are the hospitals included in the study, please mentions the name of hospitals.

5.How do you calculate sample size? Is it sufficient to draw conclusion regarding bacteremia (causative bacterial pathogens) in Indonesia?

6.What is your inclusion and exclusion criteria? Please mention Clearly.

7.Please give the ethical approval committee name and approval number and date.

8.How do assure the Quality controls and quality check of your results, either BD

135 Phoenix (Becton Dickinson) or VITEK 2 (bioMérieux, Inc., Durham, North Carolina), System?

9.What is the volume of blood sample collected and used in culture from children and adults?

10.It is better to give numerator value after percentage values.

11.Please give the full name of bacteria initially such as Staphylococcus aureus and then short form S. aureus and other bacteria throughout the manuscript.

12.Please mention the more information on infections with dengue virus and bacteremia in Indonesia.

13.Please corelate conclusion with your findings.

6. PLOS authors have the option to publish the peer review history of their article (what does this mean?). If published, this will include your full peer review and any attached files.

Reviewer #1: **Yes: **Dr Shishir Gokhale

Reviewer #2: No

---

## [Author Response · Author response to Decision Letter 0]

14 Jul 2022

Dear Dwij Raj Bhatta, PhD

Academic Editor

PLOS ONE

Thank you very much for the constructive comments and suggestions provided by the reviewers. We have carefully revised the manuscript following the suggestions. Please see the response to each comment/suggestion below.

Reviewer #1: This compilation of data from different centers over many years is commendable. This highlights the issues faced in diagnostic microbiology in developing countries. It is an interesting paper with important observations and discussions. Some spellings need review and correction. Recommend to submit after corrections.

Response: Thank you very much for your comments, we really appreciate it. 

We have corrected the spelling errors.

Reviewer #2: The Characteristics of Bacteremia among Patients with Acute Febrile Illness Requiring Hospitalization in Indonesia.

Evaluation. This report addresses an important subject in Bacteriemia and Acute Febrile illness; i.e., the worrying trend of antimicrobial resistance in bacterial pathogens (Salmonella and Non Salmonella spp) . It reports the frequency and distribution of bacterial pathogens in blood culture and its susceptibility pattern isolated from various specimens from a seven medical center in Indonesia, from which similar reports are scarce. Though it is better attempt by Soedarmono et al., to know information on bacteremia and other causative agent of Acute Febrile illeness in Indonesia.

Response: Thank you very much for your comments, we really appreciate it. 

Comments

1.Give rationale of the study? Why is NS1 antigen screening only performed? What about other viral agents related AFI?

Response: We have added more information regarding this issue in the Methods.

Lines 135-146 now read: During the baseline visit, blood was collected for cultures, clinically relevant rapid diagnostic tests when available, and dengue virus rapid diagnostic tests. Dengue virus infection remains a significant burden across Indonesia [28,29], with disease incidence increasing in recent years [30]. Though other viral agents are present in Indonesia, none are as prevalent as dengue virus [24,31], and most are challenging to diagnose due to limitations with available rapid diagnostic tests [32,33]. Given the widespread prevalence of dengue virus infection, and the very high specificity (almost 100%) and good sensitivity (70-87%) of NS1 antigen rapid diagnostic tests [34], we employed universal dengue virus screening to rapidly resolve the unknown etiologies of study participants. Participants with negative NS1 antigen tests were further considered for BSIs through blood culture tests and other etiologies, as determined through advanced testing at the INA-RESPOND reference laboratory.

2.Why you performed Blood culture 0f 1459 Cases? You have mentioned 1464 were enrolled? What about 5?? 

Response: We only performed blood culture for 1459 patients, as the remaining 5 subjects did not have enough blood for blood culture test. 

Lines 207-210 now read: The remaining 5 participants had insufficient blood specimens for following reasons: 1 adult was in a severe condition (decreased of consciousness), 2 participants (1 child and 1 adult) self-discharged against medical advice, and the guardians of 2 children refused to allow more blood to be drawn. 

3.At the end of introduction, please give some update of Acute Febrile illness and their epidemiology in Indonesia.

Response: Thank you very much for the suggestion. We have added some update of acute febrile illness and their epidemiology in Indonesia.

Lines 97-111 now read: The epidemiology of pathogens associated with fever in Indonesia is not well understood, as public health surveillance data is limited and only a few local studies have been conducted [19,21–26]. Among published studies, dengue virus, chikungunya virus, influenza virus, Salmonella Typhi, Rickettsia spp., and Leptospira spp. are consistently the most common causes of acute febrile illness hospitalizations. A study in Papua from November 1997 to February 2000 enrolled 226 hospitalized patients that were negative for malaria, the majority of whom were determined to have typhoid fever (18%), leptospirosis (12%), rickettsioses (8%), and dengue fever (7%) [23]. An observational fever study in Bandung identified dengue virus in 12.4% of fever episodes, followed by S. Typhi (7.4%), and chikungunya virus (7.1%) [24,26,27]. A 2005-2006 study in Semarang found rickettsioses and leptospirosis in 7% and 10%, respectively, of 137 acute undifferentiated fever cases [21]. The parent study of the research presented here found the most prevalent pathogens among participants at eight hospitals in 7 major cities in Indonesia to be dengue virus (27-52%), Rickettsia spp. (2-12%), S. Typhi (0.9-13%), influenza virus (2-6%), Leptospira spp. (0-5%), and chikungunya virus (0-4%) [19].

4.Which are the hospitals included in the study, please mentions the name of hospitals.

Response: We have included the name of hospitals in the Methods.

Lines 121-127 now read: A prospective observational study enrolling febrile patients who required hospitalization was conducted by the Indonesia Research Partnership on Infectious Disease (INA-RESPOND) from July 2013 to June 2016 at eight major hospitals in seven provincial capitals in Indonesia: Dr. Cipto Mangunkusumo Hospital in Jakarta, Sulianti Saroso Infectious Disease Hospital in Jakarta, Dr. Wahidin Sudirohusodo Hospital in Makassar, Dr. Sardjito Hospital in Yogyakarta, Dr. Hasan Sadikin Hospital in Bandung, Sanglah General Hospital in Denpasar, Dr. Soetomo Hospital in Surabaya, and Dr. Kariadi Hospital, in Semarang. 

5.How do you calculate sample size? Is it sufficient to draw conclusion regarding bacteremia (causative bacterial pathogens) in Indonesia?

Response: As this study was an observational study to find etiologies of acute febrile illness during a certain period of time (2013-2016), we did not specifically calculate the sample size for drawing the conclusion regarding bacteremia in Indonesia. Since we performed the analysis of blood culture results from almost all participants (>99% participants, approximately 100 adults and 100 children from each hospital), though cannot be generalizable to the Indonesian population at-large, we expected that the data will provide better understanding of the bacteremia in hospitalized population with fever and hopefully will lead to a reduction in mortality from BSIs.

6.What is your inclusion and exclusion criteria? Please mention Clearly.

Response: We have added the inclusion and exclusion criteria.

Lines 128-131 now read: Briefly, inclusion criteria consisted of axillary body temperature �38˚C, �1 year of age, and hospitalization within the past 24 hours. Patients were excluded from the study if they had subjective fever for �14 days or were hospitalized in the last 3 months.

7.Please give the ethical approval committee name and approval number and date.

Response: The name of the ethical approval committee and approval number had already provided under the “Ethical Clearance” (lines 197-203); and we have added the date.

Ethical approvals for the AFIRE study were granted by the Institutional Review Boards of the National Institute of Health Research and Development (NIHRD), Indonesia Ministry of Health (KE.01.05/EC/407/2012) dated 23 May 2012, the Faculty of Medicine at the University of Indonesia and RSUPN Dr. Cipto Mangunkusumo Hospital (451/PT02.FK/ETIK/2012) dated 23 July 2012, and RSUD Dr. Soetomo Hospital (192/Panke.KKE/VIII/2012) dated 13 August 2012.

8.How do assure the Quality controls and quality check of your results, either BD

135 Phoenix (Becton Dickinson) or VITEK 2 (bioMérieux, Inc., Durham, North Carolina), System?

Response: Blood culture tests were performed at the hospital's accredited clinical laboratory, which provides patient diagnostic services. All instruments and standards were calibrated appropriately according to manufacturer guidelines. Every site’s laboratory performed quality control (QC) to ensure proper performance and sent the QC report to protocol team to be reviewed. All tests were run alongside appropriate positive and negative control to ensure the integrity and accuracy of the results. For example, QC for VITEK 2 system; each new lot number of ID cards is tested with stock culture organisms. Susceptibility cards are tested weekly against stock culture organisms.

The QC organisms uses as follows:

Weekly: 

AST-GP 67 cards 

 Enterococcus faecalis ATCC 29212

AST-GN 66 cards

 E. coli ATCC 25922 non fermenter

 PSA ATCC 27853 fermenter

 E. coli ATCC 35218 non fermenter

ID-NH cards 

 Elkenella corrodens ATCC BAA-1152

New Lots:

ID-GP cards

 Enterococcus casseliflavis ATCC 700327

ID-GN cards

 Enterobacter hormechei (E.cloacae) ATCC 700323

Lines 163-171 now read: Blood cultures were performed and analyzed at the hospitals’ nationally accredited clinical laboratories by trained, certified staff. All instruments and standards were calibrated appropriately according to manufacturer guidelines, and all tests were run alongside appropriate positive and negative control to ensure the integrity and accuracy of the results. Organism identification was considered acceptable when the confidence level in the automated growth identification system was ≥95% probability [34]. Quality control tests were performed weekly at all site laboratories, and each new lot of ID cards was tested using validated stocks of culture organisms.

9.What is the volume of blood sample collected and used in culture from children and adults?

Response: This is already stated in the text. Blood volumes of approximately 5-8 mL for adults and 1-3 mL for children were collected from each arm, whenever possible, directly into separate aerobic blood culture bottles (lines 150-152).

10.It is better to give numerator value after percentage values.

Response: We have changed the presentation throughout the manuscript.

11.Please give the full name of bacteria initially such as Staphylococcus aureus and then short form S. aureus and other bacteria throughout the manuscript.

Response: We have followed your suggestion.

12.Please mention the more information on infections with dengue virus and bacteremia in Indonesia.

Response: We found no dengue virus and bacteremia co-infection in our study, as mentioned in the Discussion. We have added more informations about dengue virus and bacteremia.

Lines 355-368 now read: Data on co-infections with dengue virus and bacteremia is limited. A literature review of published case reports and studies from January 1943 to March 2016 found 3 studies in Singapore and Taiwan reporting concurrent bacteremia in 0.18-7% of dengue fever cases [40–42]. A concurrent dengue virus and S. Typhi case was also reported from Bandung, Indonesia [43]. In all of these studies, blood was collected for bacterial culture because patients did not improve clinically a few days to a week after dengue fever was diagnosed. Furthermore, in the majority of cases, dengue virus infection was confirmed by serology only (IgM detected or four-fold IgG increase). These reports support our finding that simultaneous infection with bacteria and dengue virus is rare. In our study, bacterial growth observed in 14 participants with positive dengue NS1 antigen tests were considered false positive blood cultures (5 Staphylococcus hominis, 4 Staphylococcus epidermidis, 1 Kocuria rosea, 1 Micrococcus aureus, 1 Staphylococcus arlettae, 1 coagulase-negative Staphylococcus spp., and 1 Staphylococcus waneri).

13.Please corelate conclusion with your findings.

Response: Thank you very much, we have correlated our conclusion with our findings.

Lines 522-541 now read: We presented aerobic blood culture findings from a multi-centre study of patients with acute febrile illness admitted to eight major hospitals across Indonesia. Our universal use of aerobic blood cultures is unique in Indonesia, the results of which help clarify the epidemiology and burden of BSI, rates of contamination among CAI, and common AMR patterns in Indonesia. Bacteremia was observed in 8.9% participants, with the most frequent pathogens being Salmonella spp., E. coli, and S. aureus. Two Salmonella spp. cases had evidence of AMR, and several E. coli cases were multidrug resistant (42.9%) or monoresistant (14.3%). Culture contamination was observed in 3.6% cases. Our data suggest that blood cultures should be included as a routine diagnostic test, and pre-screening patients for the most common viral infections, such as dengue, influenza and chikungunya viruses, would conserve scarce resources without negatively impacting patient benefit. The routine practice of AMR susceptibility testing on positive blood cultures in Indonesia is encouraging and should be continued to inform clinical decisions on patient treatment in real-time. The country could benefit from clear guidance at the national level, particularly regarding the timing of blood collection prior to antibiotic administration, the prioritization of patients with comorbidities, blood collection practices to reduce environmental contamination, and the supplementation of blood cultures with molecular assays to combat false-negative results. Additionally, Indonesia could greatly benefit from a nationwide program for the systematic collection and dissemination of blood culture and AMR results.

---

## [Editor Report · Decision Letter 1]

21 Jul 2022

PONE-D-22-07517R1The Characteristics of Bacteremia among Patients with Acute Febrile Illness Requiring Hospitalization in IndonesiaPLOS ONE

Dear Dr. Kosasih,

Thank you for submitting your manuscript to PLOS ONE. After careful consideration, we feel that it has merit but does not fully meet PLOS ONE’s publication criteria as it currently stands. Therefore, we invite you to submit a revised version of the manuscript that addresses the points raised during the review process.

We look forward to receiving your revised manuscript.

Kind regards,

Dwij Raj Bhatta, PhD

Academic Editor

PLOS ONE

Journal Requirements:

Additional Editor Comments:

manuscript requires few corrections in the section abstract method result discussion and conclusion, minor revision required . Please go through the comments made in manuscript. Serovar names of salmonella isolates obtained during present study be mentioned whereever required in the manuscript

---

## [Author Response · Author response to Decision Letter 1]

8 Aug 2022

Journal Requirements:

Response: We have reviewed our reference list to ensure that it is complete and correct. As per today (27 July 2022), no cited papers have been retracted. Therefore, no change of the reference list has been made.

Additional Editor Comments:

manuscript requires few corrections in the section abstract method result discussion and conclusion, minor revision required . Please go through the comments made in manuscript. Serovar names of salmonella isolates obtained during present study be mentioned whereever required in the manuscript

Response: Thank you very much. We have gone through the comments made in manuscript and revised the manuscript accordingly. 

When required/appropriate, we have reported names of salmonella isolates obtained during the present study throughout the manuscript (41 cases of S. Typhi and 10 cases of S. Paratyphi A). However, we have not mentioned the serovar names when we refer the subjects with Salmonella spp. and non-Salmonella spp. as groups. 

Abstract (lines 55-61 now read): Bacteremia was observed in 8.9% (92) participants, with the most frequent pathogens being Salmonella enterica serovar Typhi (41) and Paratyphi A (10), Escherichia coli (14), and Staphylococcus aureus (10). Two S. Paratyphi A cases had evidence of AMR, and several E. coli cases were multidrug resistant (42.9%, 6/14) or monoresistant (14.3%, 2/14). Culture contamination was observed in 3.6% (37) cases. Molecular and serological assays identified etiological agent in participants having negative cultures, with 23.1% to 90% of cases being missed by blood cultures.

Other changes have been made with “track changes” in the manuscript.

---

## [Editor Report · Decision Letter 2]

9 Aug 2022

The Characteristics of Bacteremia among Patients with Acute Febrile Illness Requiring Hospitalization in Indonesia

PONE-D-22-07517R2

Dear Dr. Kosasih,

We’re pleased to inform you that your manuscript has been judged scientifically suitable for publication and will be formally accepted for publication once it meets all outstanding technical requirements.

Kind regards,

Dwij Raj Bhatta, PhD

Academic Editor

PLOS ONE
---

## [Editor Report · Acceptance letter]

29 Aug 2022

PONE-D-22-07517R2 

The Characteristics of Bacteremia among Patients with Acute Febrile Illness Requiring Hospitalization in Indonesia 

Dear Dr. Kosasih:

I'm pleased to inform you that your manuscript has been deemed suitable for publication in PLOS ONE. Congratulations! Your manuscript is now with our production department. 

Kind regards, 

on behalf of

Professor Dwij Raj Bhatta 

Academic Editor

PLOS ONE